# Tandem Structures Semiconductors Based on TiO_2__SnO_2_ and ZnO_SnO_2_ for Photocatalytic Organic Pollutant Removal

**DOI:** 10.3390/nano11010200

**Published:** 2021-01-14

**Authors:** Alexandru Enesca, Luminita Isac

**Affiliations:** 1Product Design, Mechatronics and Environmental Department, Transilvania University of Brasov, Eroilor 29 Street, 35000 Brasov, Romania; 2Renewable Energy Systems and Recycling Research Center, Transilvania University of Brasov, Eroilor 29 Street, 35000 Brasov, Romania; isac.luminita@unitbv.ro

**Keywords:** metal oxides, doctor blade, tandem structures, photocatalysis, kinetics

## Abstract

The photocatalyst materials correlation with the radiation scenario and pollutant molecules can have a significant influence on the overall photocatalytic efficiency. This work aims to outline the significance of optimizing the components mass ratio into a tandem structure in order to increase the photocatalytic activity toward pollutant removal. ZnO_SnO_2_ and TiO_2__SnO_2_ tandem structures were obtained by the doctor blade technique using different mass ratios between the components. The samples contain metal oxides with crystalline structures and the morphology is influenced by the main component. The photocatalytic activity was tested using three radiation scenarios (UV, UV-Vis, and Vis) and two pollutant molecules (tartrazine and acetamiprid). The results indicate that the photocatalytic activity of the tandem structures is influenced by the radiation wavelength and pollutant molecule. The TiO_2__SnO_2_ exhibit 90% photocatalytic efficiency under UV radiation in the presence of tartrazine, while ZnO_SnO_2_ exhibit 73% photocatalytic efficiency in the same experimental conditions. The kinetic evaluation indicate that ZnO_SnO_2_ (2:1) have a higher reaction rate comparing with TiO_2__SnO_2_ (1:2) under UV radiation in the presence of acetamiprid.

## 1. Introduction

The semiconductor-mediated photocatalysis is considered as a promising pathway of removing various pollutants from aqueous and gaseous phase by directly harvesting and utilizing the solar energy [1,2,3]. Merging the sustainability with durability may be the key of transferring the photocatalytic process from the laboratory scale up to large applications. Until now there are many wide gap oxides (TiO_2_ [4,5], SnO_2_ [6,7], and ZnO [8,9]) and narrow band gap materials (Bi_2_WO_6_ [10,11], Ag_3_PO_4_ [12,13], BiPO_4_ [14,15], g-C_3_N_4_ [16,17], WO_3_ [18,19], and BiOX [20,21]) studied for the potocatalytic removal of wastewater organic contaminants and indoor pollutants. The mono-components photocatalyst have disadvantages such as narrow visible light absorption [22,23], low specific surface area [24,25], and fast charge carriers recombination [26,27].

Advanced oxidation processes (AOPs) are considered as future alternative to traditional methods of removing organic pollutants: pharmaceutical active compounds [28,29], pesticides [30,31], dyes [32,33], volatile organic compounds [34,35], etc. The sustainability represents an important advantage of AOPs due to the use of light radiation as the main energy source to provide oxidative species responsible for pollutant mineralization [36,37,38]. However, the transfer from a laboratory scale to a large application requires important optimizations in terms of energy consumption, materials, and design [39,40]. The lack of standardizations and procedures make it difficult to compare the experimental results reported in the field of photocatalysis. However, it must be underlined that the photocatalytic efficiency depends on many parameters such as pollutant type and concentration, photocatalyst composition, structure and dosage, as well as radiation wavelength and photon flux. Bi_2_MoO_6_/Fe_3_O [41] and Ta_3_B_2_@Ta_2_O_5_ [42] heterostructures were employed to investigate methylene blue (MB) dye removal under Vis irradiation. The results indicate that Bi_2_MoO_6_/Fe_3_O_4_ heterostructure have 93% photocatalytic efficiency after 180 min of irradiation with 500 W light source. Using the same irradiation scenario, the Ta_3_B_2_@Ta_2_O_5_ heterostructures exhibit 80% photocatalytic efficiency. A similar experiment was done on ZnAl_2_O_4_/Bi_2_MoO_6_ [43] photocatalyst but with a different irradiation scenario (UV light, 100 W), and after 180 min, the MB removal efficiency was 86%. Coupling the photocatalytic process with adsorption represents another way to optimize the energy consumption and to increase the pollutant removal efficiency [44].

This paper presents the correlation between the components mass ratio in a tandem structure and the photocatalytic activity using different radiation scenarios and pollutant molecules. Four tandem structures based on ZnO_SnO_2_ and TiO_2__SnO_2_ were tested using UV, UV-Vis, and Vis radiations. The tandem structures based on ZnO, TiO_2_, and SnO_2_ may benefit from the extended light absorption spectra due to the effective band gap established between the components. Additionally, due to their band energies values and efficient charge carrier’s separation, the tandem structures are able to exhibits higher performance compare with the individual components [45,46,47]. The photocatalytic properties were tested using two types of pollutants: pesticide (acetamiprid) and dye (tartrazine). Pesticides and dyes represent two important categories of organic compounds affecting the water properties and consequently the life quality. The water contamination with dyes substances such as tartrazine (Tr) have raised human health issues and this molecule is characterized by strong chemical stability toward traditional wastewater treatment processes due to the aromatic structure [48,49]. The acetamiprid pesticide is considered harmful due to the high toxicity (especially at high concentrations), high accumulation rate, and possible carcinogenic effect induced by their non-biodegradable aromatic structure [50,51]. The results correlate the tandem structure composition with the photocatalytic kinetics based on radiation parameters (wavelength, total irradiance, and photon flux) for each pollutant.

## 2. Materials and Methods

### 2.1. Tandem Structures Films Based on Metal Oxides

The ZnO, SnO_2_, and TiO_2_ metal oxide powders were purchased (Sigma Aldrich, Saint Louis, MO, USA) and used without any purification procedures. Four samples with different mass ratio composition were prepared as follows:(1)Sample ZnO_SnO_2_ (2:1) with a mass ratio between ZnO and SnO_2_ of 2:1;(2)Sample ZnO_SnO_2_ (1:2) with a mass ratio between ZnO and SnO_2_ of 1:2;(3)Sample TiO_2__SnO_2_ (2:1) with a mass ratio between TiO_2_ and SnO_2_ of 2:1;(4)Sample TiO_2__SnO_2_ (1:2) with a mass ratio between TiO_2_ and SnO_2_ of 1:2.

Another three samples containing bare ZnO, SnO_2_, and TiO_2_ were prepared to be compared with tandem samples.

The deposition technique was a doctor blade and the substrate was microscopic glass. In the first step, the metal oxide-based paste was prepared considering the above mass ration between the components. The metal oxide powder was dispersed into a mixture of ethanol, acetylacetonate, triton ×100 in a volumetric ratio 10:1.5:1.5. The dispersion procedure includes 30 min of vigorous magnetic stirring to assure the paste uniformity. In the second step, the substrate, previously cleaned using surfactants to remove grease traces and then immersed in ethanol for 20 min using an ultrasound bath, was immobilized on a flat surface using a non-conductive transparent tape. The third step is represented by the paste addition (100 μL) on the substrate surface where a glass scraper ensures the uniform paste distribution at constant velocity (1.5 s/cm^2^). The last step includes a thermal treatment procedure done at 500 °C for 5 h in order to eliminate the organic additives.

### 2.2. Photocatalytic Procedures

The photocatalytic experiments where done in a reactor able to assure a uniform light intensity distribution due to 3 light sources placed in suitable positions. The reactor room is characterized by low humidity and 20–25 °C temperature (depending on the radiation sources). Three light scenarios where employed and the corresponding total irradiance was measured and presented in Table 1. The UV irradiation was provided by 18 W black light Philips tubes T8 model (Amsterdam, Olanda), with 3Lx flux intensity, spectral range between 310 and 390 nm, and a maximum emission at 365 nm. The Vis irradiation was obtained from 18 W white cold light Philips tubes TL-D 80/865 model, with 28Lx flux intensity, spectral range between 400 and 700 nm, and a maximum emission at 565 nm.

The irradiance characteristics considering each light source type were *E_UV_* = 2.9 W/m^2^ corresponding to the UV light and *E_Vis_* = 4.2 W/m^2^ corresponding to the Vis light. The maximum photon flux, Φ, was calculated using Equation (1) and the values are presented in Table 1. The evaluation takes into consideration the light maximum wavelength (*λ_UV_*_, max_, *λ_Vis_*_, max_) as well as the number of irradiation sources (*n_uv_* and, respectively, *n_vis_*) [52].
(1)$$\Phi =\frac{E_{UV}\cdot \lambda_{UV}\cdot n_{UV}+E_{Vis}\cdot \lambda_{Vis}\cdot n_{Vis}}{h\cdot c\cdot N_{Av}^{}},$$
where: the Planck constant (*h*), the speed of light (*c*), and the Avogadro number have the usual values.

Two organic pollutants were used to evaluate the tandem structures photocatalytic properties: acetamiprid pesticide (Apd) and tartrazine dye (Tr). During the investigation, the tandem structures were immersed for 10 h in 35 mL of pollutant solution (0.025 mM). In the first 120 min, the tandem structures were kept in the dark which is enough to attempt the absorption equilibrium. During the following 8 h, the tandem structures were irradiated based on the three radiation scenarios presented in Table 1. The changes in pollutant concentration were investigated based on the UV-Vis calibration curve and hourly evaluated up to 8 h of photo-catalysis.

The photocatalytic removal efficiency was calculated using Equation (2):(2)$$\eta =\left[\frac{\left(C_{0}-C\right)}{C_{0}}\right]\cdot 100,$$
where: *C*_0_ represents the initial concentration and *C* represents the pollutant concentration at moment t. The UV-Vis calibration curve based on the absorption spectra of the pollutant was done using the following procedure: (1) several solutions with accurately known concentrations (in the range of working conditions) were prepared; (2) the absorbance at the wavelength of strongest absorption was measured; (3) a graph plot representing the absorbance against concentration was done considering Beer-Lambert law for diluted solutions.

### 2.3. Investigation Instruments

The crystalline composition was studied using X-ray diffraction (XRD, Bruker D8 Discover Diffractometer, Karlsruhe, Germany) with a setup consisting on locked-couple system at 0.004 degree scan step and 0.02 s/step. Field emission scanning electron microscopy (FESEM, SU8010, Fukuoka, Japan) was used to investigate the samples morphology, operated at an accelerated voltage of 25 kV. The optical and photocatalytic investigations were done using the UV-Vis spectrometry (Perkin Elmer Lambda 950, Waltham, MA, USA) technique with a scanning step of 1.0 nm and 6° incidence angle for reflectance measurements. Total irradiance for each scenario was measured using a class A pyranometer (SR11, Hukselflux, Berlin, Germany) and the sensor was placed in the central position of the sample holder.

## 3. Results and Discussion

### 3.1. Composition and Morphology

The diffraction analysis presented in Figure 1 indicates the presence of crystalline structures in all samples, which is a pre-requisite for further photocatalytic applications [53,54]. The peak intensity varies based on the mass ration of each component. There are no additional peaks which may suggest the formation of other non-stoichiometric metal oxides or carbonaceous species [55]. However, possible doping between metal oxides during the post-deposition thermal treatment cannot be excluded. Samples ZnO_SnO_2_ contains ZnO with hexagonal crystalline structure (ICCD 89-1397) and SnO_2_ with tetragonal structure (ICCD 41-1445) which is consistent with the as-received powders. The ZnO peak intensity increases in sample ZnO_SnO_2_ (2:1) where the ZnO ratio is double compared with SnO_2_. A similar observation can be done for ZnO_SnO_2_ (1:2) where the SnO_2_ peak is predominant.

The samples TiO_2__SnO_2_ have the characteristic peaks of anatase TiO_2_ (ICCD 89-4203) and SnO_2_ tetragonal phase. However, there are no peaks corresponding to rutile TiO_2_ even if the TiO_2_ Degussa contains both crystalline structures. Based on the close proximity of the metal oxide diffraction peaks, the rutile TiO_2_ main peak may be covered by the SnO_2_ (110) peak intensity which is present in the same diffraction area [56]. The SnO_2_ peak intensity increases in sample TiO_2__SnO_2_ (1:2) where the SnO_2_ ratio is higher. Moreover, the shape of the SnO_2_ peaks is different compared with ZnO_SnO_2_ samples that may suggest changes of the crystallite sizes [57,58]. The post-deposition thermal treatment used in order to eliminate all the carbonaceous species may also influence the synergy between the metal oxide particles.

The EDS measurements were done to investigate the elemental composition at the tandem structures surface and the results are presented in Table 2. The qualitative results were compared with theoretical oxygen content calculated considering the stoichiometric compounds identified in XRD results. In all samples, the values indicate an oxygen excess which is consistent with our previous studies [59,60] showing that the samples submitted to post-deposition treatment in reach oxygen atmosphere will develop higher oxygen content. The ratio between metal ions at the surface is not the same as the initial values used during the tandem structure deposition. This was expected considering that the deposition method and the dispersive procedure do not allow an accurate control in terms of homogeneity [61]. In this case, the possibility of composition variation may occur especially in bulk where the tendency of forming aggregates is higher. However, the element which is in higher ratio during the deposition will remain predominant at the sample surface. In addition, due to the annealing treatment in air at elevated temperatures, most of the oxygen vacancies will be passivated according with Equation (3).
V_O_^−^ + 1/2O_2_ → O_O_*^x^* + 2h^−^.(3)

The morphology plays an important role in the photocatalytic activity considering that most of the active sites that generate the oxidative species are located on the film surface [62,63]. The results were correlated with the quantitative evaluation of the tandem structures presented in Table 3.

The SEM images (Figure 2) indicate that the surface morphology depends on the composition ratio of the tandem structure. The mono-component samples (Figure 2a–c) exhibit lower thickness compared to tandem systems due to the uniform particles size (60–80 nm for ZnO, 50–70 nm for SnO_2_, and 20–40 nm for TiO_2_) used to prepare the precursor paste. The aggregates formation is present in all samples but the size is lower in mono-component samples compared with the tandem system. The tandem systems include particles with various sizes which have the tendency to form larger agglomerations which are not completely dispersed during the diffusion process. Sample ZnO_SnO_2_ (2:1) (Figure 2d) exhibits higher density compared with sample ZnO_SnO_2_ (1:2) and large grains with irregular shape. When SnO_2_ is the majority component, the sample density increases as well as the film thickness. Sample ZnO_SnO_2_ (1:2) shows a porous morphology and a thickness of around 3.52 μm. However, the tandem sample composed from TiO_2_ and SnO_2_ presents small grains closely packed and smaller thickness values which indicate a good structural compatibility between the components. The formation of surface irregularities on the TiO_2__SnO_2_ samples may be attributed to the residual particles that occur during the post-deposition thermal treatment [64,65]. The extension of these irregularities is limited, indicating that it is not a characteristic of the tandem structures morphology. The organic additives were used to increase the mechanical adhesion on the microscopic glass substrate, otherwise there is a risk of film collapse during the photocatalytic experiments [66].

### 3.2. Photocatalytic Activity

#### 3.2.1. Photocatalytic Efficiencies and Kinetics

The photocatalytic removal efficiency of Tr and Apd molecules were tested for all tandem structures using three different irradiation scenarios. The lowest photocatalytic efficiencies (Figure 3a,b) were obtained using a Vis irradiation scenario due to the metals oxides band gap energies which correspond to the UV region.

The bare ZnO and TiO_2_ samples exhibit 58% and, respectively, 76% photocatalytic removal efficiencies under UV irradiation. The lowest photocatalytic activity was recorded for bare SnO_2_ which is able to reach only 35% efficiency using UV light sources. The tandem structures show higher photocatalytic activity due to the cumulated effect of multiple charge carriers generation and low recombination rate. The highest efficiency of 90% was recorded for Tr removal during the UV irradiation of TiO_2__SnO_2_ (2:1) sample (Figure 3b). Using the same irradiation scenario but changing the TiO_2__SnO_2_ ratio to 1:2, the photocatalytic removal efficiency decreased to 80%. These results indicate that TiO_2_ content in the TiO_2__SnO_2_ tandem structure is the driving photocatalitic factor with a significant contribution on the pollutant removal efficiency.

The ZnO_SnO_2_ samples exhibit similar photocatalytic behavior (Figure 3a) and reach 73% photocatalytic efficiency under UV irradiation when the ZnO_SnO_2_ ratio is 2:1. The total irradiance (Figure 3c) and the photon flux (Figure 3d) evaluations indicate that in order to obtain high photocatalytic values the wavelength radiation must be correlated with the photocatalytic materials. Otherwise, high irradiance and photon flux is not enough to enhance the photocatalytic properties. By coupling UV with Vis radiation, the photocatalytic removal efficiency decreases due to the lower photons concentration available to generate the oxidative species during the Tr removal.

The Apd photocatalytic removal efficiency (Figure 4a,b) is lower compared with Tr due to the higher chemical stability of the Apd molecule [67].

The highest photocatalytic efficiency (Figure 4b) was recorded for sample TiO_2__SnO_2_ (2:1) under UV irradiation able to reach 57% compared with 42% for bare TiO_2_ or 30% for bare ZnO. However, the difference between TiO_2__SnO_2_ and ZnO_SnO_2_ samples in terms of photocatalytic activity is not so obvious for the Apd molecule. Due to the higher induction period, the samples require a longer period to produce enough oxidative species (manly ^·^O_2_^−^, HO^−^ radicals) necessary to decompose the pollutant molecules. Figure 4c,d indicates that using 12.6 W/cm^2^ total irradiance and 24.83 μmol/m^2^·s photo flux from UV sources can induce an increase of the photocatalytic activity compared with 17.3 W/cm^2^ and 68.42 μmol/m^2^·s from Vis sources. Based on the photocatalytic efficiency curve shape, the ZnO_SnO_2_ samples will reach the saturation point faster compared with TiO_2__SnO_2_ tandem structures. In this case, longer irradiation periods can increase the photocatalytic activities differences between the samples.

Furthermore, the influence of the light radiation and tandem structure composition was correlated with the photocatalytic kinetic data, based on the simplified Langmuir-Hinshelwood (L-H) mathematical equation, see Equation (4):(4)$$\ln C/C_{0}=-kt.$$

The kinetic evaluation of the Tr photocatalytic removal (Figure 5) indicates that the rate constant is almost double (Table 4) when TiO_2_/SnO_2_ samples are irradiated with UV radiation compared with mixed UV-Vis radiation. Additionally, the photocatalytic activity under UV radiation of the TiO_2__SnO_2_ samples is 28× faster compared with Vis radiation.

Lower differences are recorded for ZnO_SnO_2_ samples where the photocatalytic activity under UV radiation is 1.4× higher compared with mixed UV–Vis radiation and 11× faster compared with Vis radiation. Based on the comparative evaluation of the rate constant vs. photon flux (Figure 5c), TiO_2__SnO_2_ (2:1) exhibits the optimum photocatalytic activity for both UV and UV-Vis radiation scenarios.

The kinetic evaluation for Apd photocatalytic removal (Figure 6) indicates that the reaction rate decreases significantly compared with Tr photocatalytic removal and the differences between radiation scenarios are lower.

Even if the photocatalytic activity remains higher under UV radiation, the reaction rate is influenced by the pollutant molecule stability. The TiO_2__SnO_2_ and ZnO_SnO_2_ samples exhibit under UV radiation 2× higher photocatalytic activity compared with UV-Vis radiation. However, this results indicate that the ZnO_SnO_2_ (2:1) sample is suitable for the pesticide removal and the rate constant (Figure 6c) at 24.83 μmol/m^2^·s photo flux is higher comparative with the TiO_2__SnO_2_ (1:2) tandem structure.

The influence of photon absorption based on the UV scenario was evaluated using the L-H model proposed by Turchi and Ollis [68]:(5)$$r=-\frac{dC}{dt}=\frac{k_{r}K_{S}C}{1+K_{S}C},$$
where *k_r_* (mol/L·min) represent the apparent reaction rate constant, *C* is the acetamiprid and tartrazine (Tr and Apd) concentrations (mol/L), *r* (mol/L·min) represent the photocatalytic removal rate, and *K_S_* (L/mol) is the apparent adsorption constant. Consequently, the *k_r_·K_S_* term is used to describe the apparent rate constant *k* (min^−1^) corresponding to the photocatalytic activity. The *k_r_* constant should consider the photon flux values and Equation (5) can be changed accordingly:(6)$$\frac{1}{r}=\frac{1}{k_{r}K_{S}}\cdot \frac{1}{C}+\frac{1}{k_{r}}.$$
Using the linear plot 1/*C* vs. 1/*r*, as well as the (1/*k_r_*) intercept and the (/*k_r_K_S_*) slope allow evaluating the kinetic parameters, and the results obtained for UV radiation are presented in Table 5.

The values based on the mathematical model indicate that L-H exhibit a good fit with the experimental data for ZnO_SnO_2_ (2:1), TiO_2__SnO_2_ (2:1), and TiO_2__SnO_2_ (1:2) tandem structures. In these three situations, the apparent reaction rates and the apparent adsorption constant have the some order of magnitude, while for the ZnO_SnO_2_ (1:2) tandem structure, there is one order of magnitude difference. These results were also observed by Isac et al. [69], showing that using tandem structures based on metal oxides with compatible band gaps will favor the charge carriers photogeneration and conversion to oxidative species. The results show that it is feasible to consider that the degradation mechanism is affected not only by the radiation type but also by the pollutant chemical stability during the photocatalytic activity.

#### 3.2.2. Photocatalytic Mechanisms

The structure representation of the tandem components band energies (Figure 7) will provide additional information regarding the photocatalytic experimental results. The charge carriers concentration and mobility depend on the suitable disposal of the energy bands and were evaluated based on the Gao et al. [70] and Mise et al. [71] algorithms. The details procedure for the conduction band (CB) and valence band (VB) potentials evaluation for tandem structure was previously presented [72].

The bands energy diagram indicates that under light irradiation the photogenerated electrons from the SnO_2_ valence band are transferred on zinc oxide valence band which is the closest energy level in the ZnO_SnO_2_ tandem structure or on titanium oxide valence band in the TiO_2__SnO_2_ sample. The photogenerated electron-hole pairs within the charged space region are efficiently separated by the electric field [73]. The charge carrier’s insertion in the depletion layer will induce an increase of the concentration gradient over the tandem structure, resulting in the development of a diffuse layer [74,75]. Consequently, due to the combined effect of drift and diffusion, the photogenerated electrons and holes will flow through the tandem components. The ZnO, TiO_2_, and SnO_2_ conduction band (CB) edges are situated at −0.25, −0.31, and −0.10 eV vs. normal hydrogen electrode (NHE). The SnO_2_ valence band (VB) edge (+3.41 eV) is lower compared to ZnO (+2.85 eV) and TiO_2_ (+3.00 eV). The charge carrier’s diffusion in both tandem structures will evolve form the SnO_2_ valence band to the most closely energy level represented by the ZnO or TiO_2_ valence band and overcome the band gap energy in order to transfer in the conduction bands [76]. The effective band gap value of the ZnO_SnO_2_ and TiO_2__SnO_2_ tandem systems was evaluated according to Scanlon et al. [77]. The results confirm the extended Vis activation (up to 420 nm) of the ZnO_SnO_2_ which explain the presence of photocatalytic activity (even at low values) under the Vis irradiation. As expected, there is no evidence that the band gap values depends on the mass ration between the components.

The pollutant mineralization is related with the tandem structure ability to generate interfacial oxidative species according to the following Equations (7)–(10):Tandem structure + hν → e^−^ + h^+^,(7)
h^+^ (Tandem structure) + H_2_O → OH (Tandem structure) + H^+^,(8)
Organic Pollutant + OH → Photocatalysis products,(9)
O_2_ + e^−^ → O_2_^−^ (aqueous solution).(10)

## 4. Conclusions

The photocatalytic activity evaluation shows that the highest efficiency (90%) for Tr dye removal was obtained for sample TiO_2__SnO_2_ (2:1) under UV radiation. Based on the same radiation scenario, the ZnO_SnO_2_ (2:1) sample exhibits 70% photocatalytic removal efficiency. The photocatalytic reaction rate is significantly influenced by the radiation type and tandem composition. The TiO_2__SnO_2_ samples reaction rate is double under UV radiation compared with UV–Vis radiation and 28× higher compared with Vis radiation. Compared with bare ZnO, SnO_2_, and TiO_2_, the tandem structures exhibit improved photocatalytic efficiency due to the lower recombination rate and higher photogenerated charge carrier density.

The changes in the photocatalytic activity are smaller during the Apd photocatalytic removal. The reaction rate for TiO_2__SnO_2_ samples is 13× higher in UV compared with Vis radiation. Additionally, the constant rate of the ZnO_SnO_2_ (2:1) sample is higher than that of TiO_2__SnO_2_ (1:2) under UV radiation. The charge carriers mobility based on the energy band diagram indicate that the photogenerated electrons follows the type II mechanism and have a significant contribution on the development of oxidative species.

These results indicate that the photocatalytic process optimization should consider a suitable optimization between the radiation sources, photon flux, catalyst materials, and pollutant type. Using high irradiance and photon flux is not enough to assure high photocatalytic efficiency. The correlations of the irradiation scenario with photocatalyst materials and pollutant type will have a positive impact on the overall photocatalytic efficiency, improving the charge carrier’s photogeneration and the formation of oxidative species.

## Figures and Tables

**Figure 1 nanomaterials-11-00200-f001:**
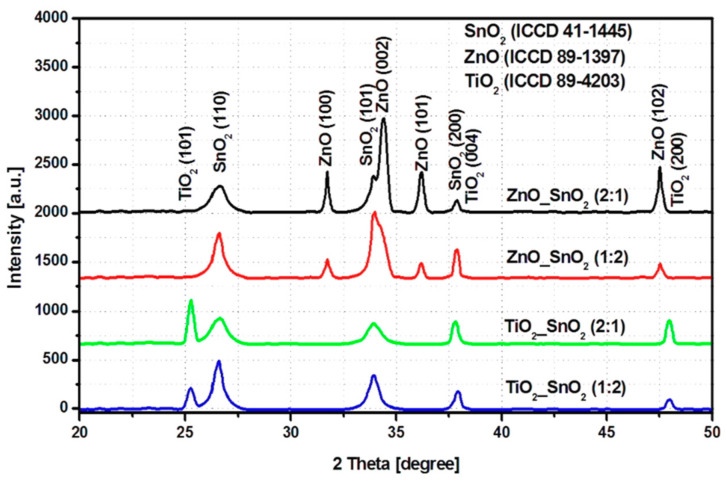
Diffraction patents for the tandem metal oxides structures.

**Figure 2 nanomaterials-11-00200-f002:**
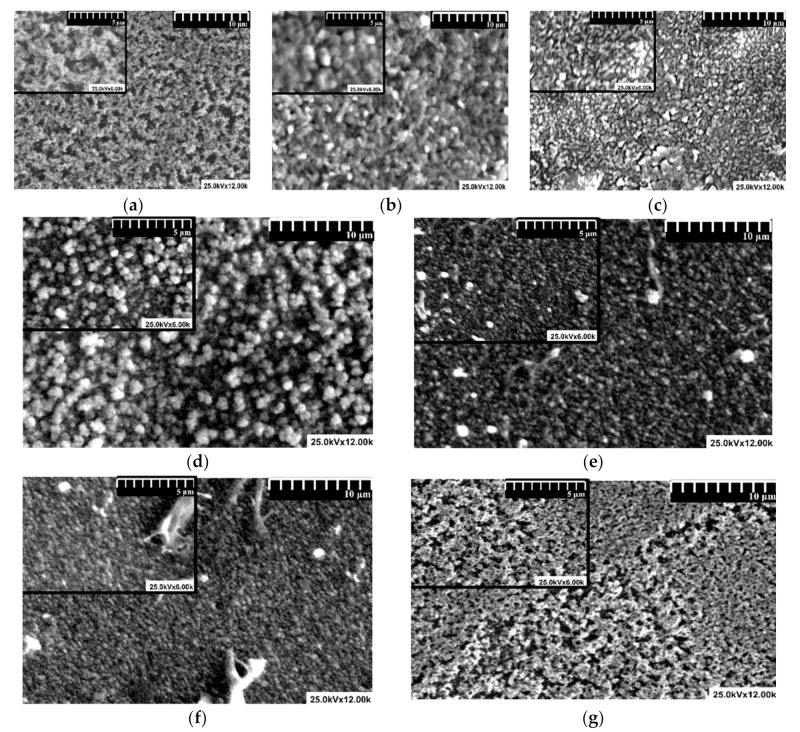
Scanning electron microscopy images of the samples: (**a**) ZnO, (**b**) SnO_2_, (**c**) TiO_2_, (**d**) ZnO_SnO_2_ (2:1), (**e**) ZnO_SnO_2_ (1:2), (**f**) TiO_2__SnO_2_ (2:1), and (**g**) TiO_2__SnO_2_ (1:2).

**Figure 3 nanomaterials-11-00200-f003:**
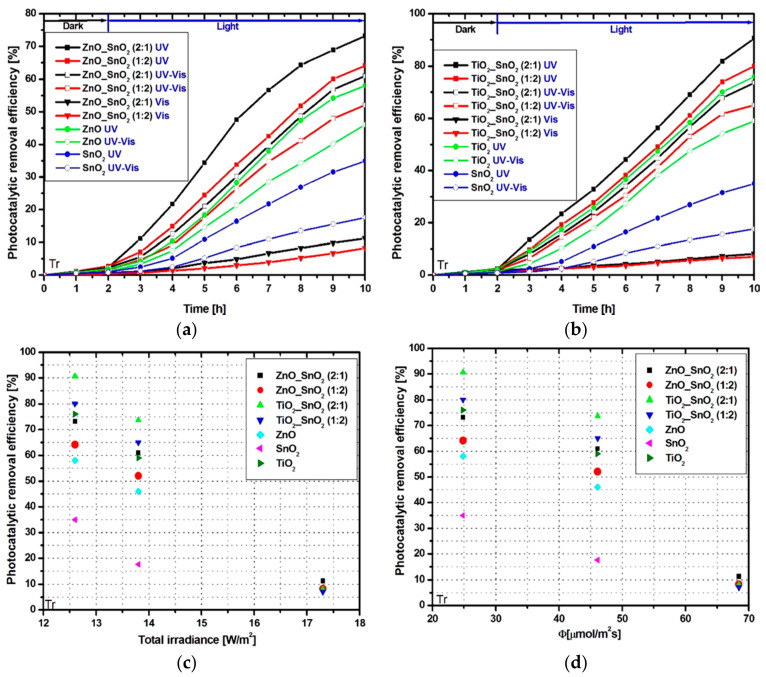
Photocatalytic activity toward Tr molecule: (**a**,**b**) removal efficiency, (**c**) photocatalytic removal efficiency vs. total irradiance, and (**d**) photocatalytic removal efficiency vs. photon flux.

**Figure 4 nanomaterials-11-00200-f004:**
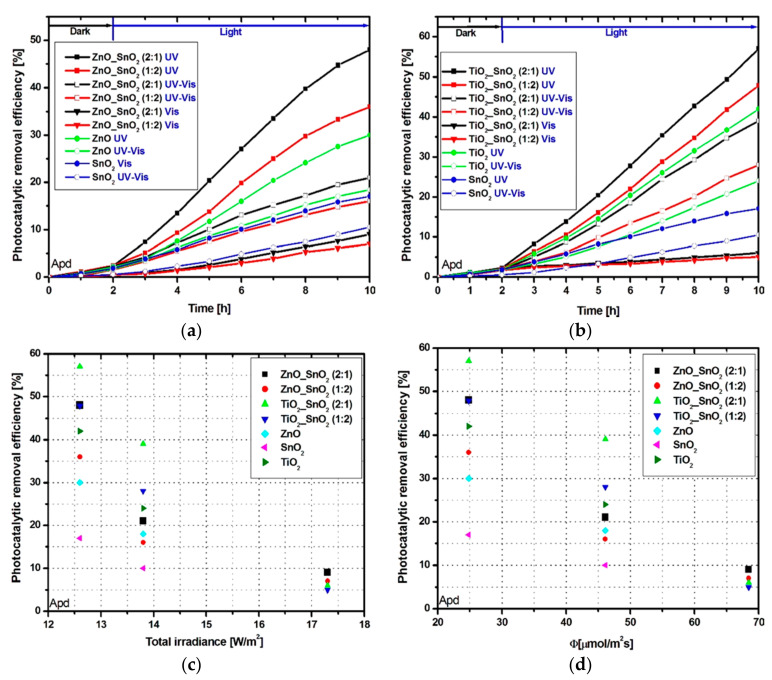
Photocatalytic activity toward Apd molecule: (**a**,**b**) removal efficiency, (**c**) total irradiance vs. pollutant removal efficiency, and (**d**) photocatalytic removal efficiency vs. photon flux.

**Figure 5 nanomaterials-11-00200-f005:**
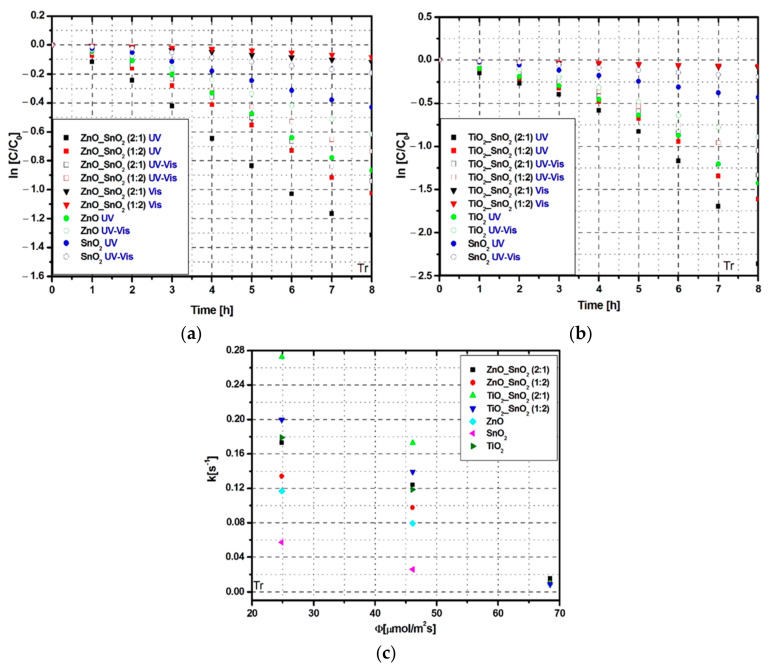
Kinetic evaluation of photocatalytic activity toward Tr molecule: (**a**,**b**) removal kinetics, (**c**) photon flux vs. removal rate constant.

**Figure 6 nanomaterials-11-00200-f006:**
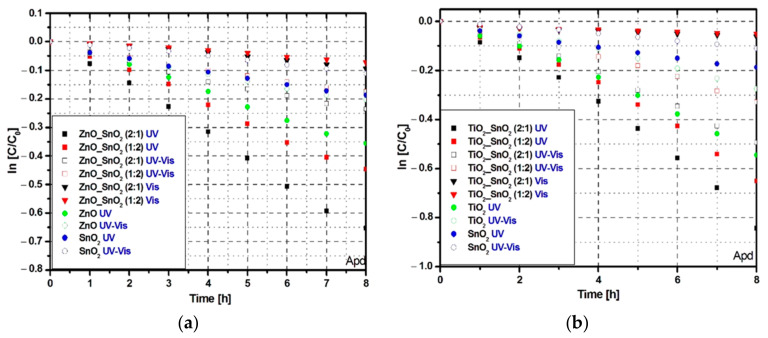
Kinetic evaluation of photocatalytic activity toward Apd molecule: (**a**,**b**) removal kinetics, (**c**) photon flux vs. removal rate constant.

**Figure 7 nanomaterials-11-00200-f007:**
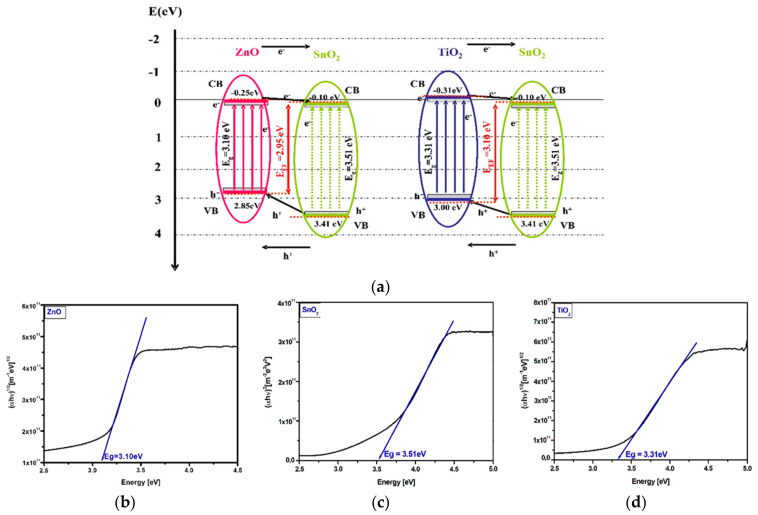
Band energy diagrams (**a**) and the corresponding band gap values (**b**–**d**).

**Table 1 nanomaterials-11-00200-t001:** Irradiation sources and total irradiance.

Irradiation Sources	UV (310–390 nm)	Vis (400–800 nm)	Total Irradiance (W/m^2^)	Φ [μmol/(m^2^·s)]
UV light	3	0	12.6	24.83
UV-Vis light	2	1	13.8	46.10
Vis light	0	3	17.3	68.42

**Table 2 nanomaterials-11-00200-t002:** Average atomic composition at the surface (EDS) and the corresponding oxygen percentage based on stoichiometric composition.

Components	Elemental Composition [% at]				
	Zn	Ti	Sn	O	O_th_ ^1^
ZnO_SnO_2_ (2:1)	28.8	-	12.6	56.3	54.0
ZnO_SnO_2_ (1:2)	13.8	-	22.5	61.7	58.8
TiO_2__SnO_2_ (2:1)	-	19.3	12.1	65.9	62.8
TiO_2__SnO_2_ (1:2)	-	10.7	21.3	66.2	64.0

^1^ Theoretic content calculated based on the stoichiometry.

**Table 3 nanomaterials-11-00200-t003:** Tandem structures quantitative evaluation.

Properties	ZnO	SnO_2_	TiO_2_	ZnO_SnO_2_ (2:1)	ZnO_SnO_2_ (1:2)	TiO_2__SnO_2_ (2:1)	TiO_2__SnO_2_ (1:2)
Thickness [μm] ^1^	2.63	2.41	1.88	3.27	3.52	2.13	2.61
Volume [cm^3^]	18.07 × 10^−5^	17.84 × 10^−5^	14.62 × 10^−5^	23.68 × 10^−5^	28.17 × 10^−5^	16.51 × 10^−5^	18.33 × 10^−5^
Density [g/cm^3^]	6.4	6.2	4.9	7.1	6.3	5.4	6.5
Weight [g]	1.15 × 10^−3^	1.10 × 10^−3^	7.16 × 10^−4^	1.68 × 10^−3^	1.77 × 10^−3^	8.91 × 10^−4^	1.19 × 10^−3^

^1^ Calculated from the reflectance spectra at 6° incident angle.

**Table 4 nanomaterials-11-00200-t004:** Kinetic data corresponding to Tr pollutant.

Kinetic Data	ZnO_SnO_2_ (2:1)		ZnO_SnO_2_ (1:2)		TiO_2__SnO_2_ (2:1)		TiO_2__SnO_2_ (1:2)		ZnO		SnO_2_		TiO_2_	
	k [s^−1^]	R^2^	k [s^−1^]	R^2^	k [s^−1^]	R^2^	k [s^−1^]	R^2^	k [s^−1^]	R^2^	k [s^−1^]	R^2^	k [s^−1^]	R^2^
Tr														
UV	0.1731	0.9972	0.1340	0.9922	0.2724	0.9495	0.1997	0.9709	0.1169	0.9885	0.0571	0.9927	0.1790	0.9771
UV-Vis	0.1240	0.9898	0.0976	0.9927	0.1728	0.9724	0.1392	0.9842	0.0794	0.0016	0.0257	0.9940	0.1187	0.9870
Vis	0.0153	0.9962	0.0104	0.9840	0.0097	0.9910	0.0086	0.9938	-	-	-	-	-	-
Adp														
UV	0.0844	0.9987	0.0582	0.9979	0.1030	0.9910	0.0804	0.9904	0.0462	0.9989	0.0228	0.9959	0.0678	0.9950
UV-Vis	0.0291	0.9954	0.0214	0.9962	0.0624	0.9949	0.0406	0.9954	0.0252	0.9968	0.0139	0.9977	0.0340	0.9938
Vis	0.0199	0.9942	0.0092	0.9943	0.0063	0.9456	0.0063	0.9456	-	-	-	-	-	-

**Table 5 nanomaterials-11-00200-t005:** Kinetic parameters based on Equation (6) for UV radiation.

Tandem Structure, Pollutant	*k_r_*·10^8^ (mol/L·min)	*K_S_* (mol/L)
ZnO/SnO_2_ (2:1), Tr	4.13	163,392.4
ZnO/SnO_2_ (2:1), Apd	2.44	105,831.8
ZnO/SnO_2_ (1:2), Tr	1.83	95,273.5
ZnO/SnO_2_ (1:2), Apd	1.38	48,527.9
TiO_2_/SnO_2_ (2:1), Tr	5.28	294,772.3
TiO_2_/SnO_2_ (2:1), Apd	3.71	149,934.0
TiO_2_/SnO_2_ (1:2), Tr	5.14	263,972.6
TiO_2_/SnO_2_ (1:2), Apd	3.53	135,729.1

## Data Availability

Data presented in this study are available by requesting from the corresponding author.

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
