# Peer review of "Tandem Structures Semiconductors Based on TiO2_SnO2 and ZnO_SnO2 for Photocatalytic Organic Pollutant Removal"

_nanomaterials, 2021, doi:10.3390/nano11010200_

Round 1

Reviewer 1 Report

Enesca et al. describe the use of tandem semicontuctors based on SnO2/TiO2 and ZnO/TiO2 for the photocatalytic removal of two compounds: acetamiprid, a pesticide, and tartrazine, a dye.

The authors investigated in a good way the composition influence of the composites based semiconductors, in addition the authors investigate the influence of the light source. 

The manuscript is well written, but some modifications are necessary:

  • First of all in my opinion the introduction is very poor, in addition several important works are omitted: ACS Omega 2020, 5, 34, 21651-21661; Catalysts 2016, 6(6), 84; Catalysts 2018, 8 (2) , 74.
  • I suggest to add as supplementary informations the spectra of the three different light sources;
  • The authors used Degussa TiO2 with both Anatase and Rutile phase, the conversion of TiO2 to Rutile phase is irreversible, the authors, during the explanation of XRD pattern, says that is no present ruthile phase. I suggesta better explaination of the XRD Figure, referee to the following manuscripts: Scientific Reports 5, Article Number:17801 (2016); Scientific Reports 5, Article Number: 20491 (2016). Maybe some diffraction peaks can overlap with each others.
  • An additional characterization of the materials can be done by Raman Spectroscopy, it is a suggestion for the authors;
  • It is well known that,if TiO2, SnO2 and ZnO are not properly doped, their band gap are in the UV region, why the authors decide to test visible light? In my opinion the light source influence has not great sense.

I suggest to accept the manuscript after major revision.

Author Response

Dear Reviewer,

We express our gratitude to your work and guidance that was helping us to improve the quality of this manuscript.

We have considered all your comments and suggestion in the new revised form of the manuscript. The changes where highlighted in red.

Enesca et al. describe the use of tandem semicontuctors based on SnO2/TiO2 and ZnO/TiO2 for the photocatalytic removal of two compounds: acetamiprid, a pesticide, and tartrazine, a dye.

The authors investigated in a good way the composition influence of the composites based semiconductors, in addition the authors investigate the influence of the light source. 

The manuscript is well written, but some modifications are necessary:

Q1. First of all in my opinion the introduction is very poor, in addition several important works are omitted: ACS Omega 2020, 5, 34, 21651-21661; Catalysts 2016, 6(6), 84; Catalysts 2018, 8 (2) , 74.

A1. Thank you for your frank opinion. We have new references and the Introduction part was improved.

Lines 46-56

Lines 60-63

Lines 65-71

Q2. I suggest to add as supplementary informations the spectra of the three different light sources;

A2. We have added supplementary infromations regarding the light sources.

Lines 101-104

Q3. The authors used Degussa TiO2 with both Anatase and Rutile phase, the conversion of TiO2 to Rutile phase is irreversible, the authors, during the explanation of XRD pattern, says that is no present ruthile phase. I suggesta better explaination of the XRD Figure, referee to the following manuscripts: Scientific Reports 5, Article Number:17801 (2016); Scientific Reports 5, Article Number: 20491 (2016). Maybe some diffraction peaks can overlap with each others.

A3. Thank you for the observation. We have improved the discussion regarding XRD patterns and added new references.

Lines 163-167

Q4. An additional characterization of the materials can be done by Raman Spectroscopy, it is a suggestion for the authors;

A4. Thank you for the suggestion. Unfortunately, we don’t have a Raman Spectroscopy setup in our R&D Institute. We are aware about the importance of Raman analysis in this field (as well as XPD). However, we try to cover some aspects of surface composition using EDX.

Q5. It is well known that,if TiO2, SnO2 and ZnO are not properly doped, their band gap are in the UV region, why the authors decide to test visible light? In my opinion the light source influence has not great sense.

A5. It is true that TiO2, SnO2 and ZnO take individually are materials which use mostly tue UV light spectra. When they are combined in tandem or heterostructures systems the optical properties may change due to the effective band gap of the type II mechanism. Scanlon was the first (from my knowledge) who made and reported this observation based on his work regarding photoactive heterostructures materials. We made several investigations on different material and the results indicate that the effective band gap may change in certain conditions (most of them depends on the tandem composition and crystalline structure). The tandem system will still absorb mostly in UV region but also can use a small part of Vis region (at the boundary between UV-Vis).

Reviewer 2 Report

The works presents the synthesis of tandem ZnO-SnO2 and TiO2-SnO2 as photocatalytic structures. The study is of interest in its field and well organized. Some details need to be addressed before further consideration:

  • The title mentions the tandem structures SnO2-TiO2 and ZnO-TiO2 while the rest indicates ZnO-SnO2 and TiO2-SnO2. Please correct.
  • Please indicate the main parameter values for the photocatalyst performance in the abstract.
  • While the Introduction mentions the drawbacks of the mono-component photocatalysts, the authors fail to highlight the benefits of the double-component ones and specifically those based on the selected oxides ZnO, TiO2 and SnO2.
  • While it is true that lack of standardization impedes a comparison with the literature, it is still of interest to show corresponding results for the chosen oxide photocatalysts. Obviously, the photocatalytic performance depends on the radiation wavelength, photon flux.
  • The authors fail in supporting the choice of the pollutant types investigated.
  • Why was Doctor Blade selected to obtain the tandem structures?
  • The section 2.1. could be compressed as the synthesis details are similar. Doctor Blade details are missing.
  • English needs revision (e.g. L79, L286-287)
  • Describe the method employed to obtain the UV-VIS calibration curve in section 2.2.
  • 2 – is it advised to show also the first layer morphology.

Author Response

Dear Reviewer,

We express our gratitude to your work and guidance that was helping us to improve the quality of this manuscript.

We have considered all your comments and suggestion in the new revised form of the manuscript. The changes where highlighted in red.

The works presents the synthesis of tandem ZnO-SnO2 and TiO2-SnO2 as photocatalytic structures. The study is of interest in its field and well organized. Some details need to be addressed before further consideration:

Q1. The title mentions the tandem structures SnO2-TiO2 and ZnO-TiO2 while the rest indicates ZnO-SnO2 and TiO2-SnO2. Please correct.

A1. Thank you for pointing this typo mistake. We have corrected the title including the tandem structure composition.

Line 3

Q2. Please indicate the main parameter values for the photocatalyst performance in the abstract.

A2. We have included the representative photocatalytic efficiencies values in the abstract part of the manuscript.

Lines 21-23

Q3. While the Introduction mentions the drawbacks of the mono-component photocatalysts, the authors fail to highlight the benefits of the double-component ones and specifically those based on the selected oxides ZnO, TiO2 and SnO2.

A3. Thank you for the remarque. We have inserted additional paragraph to explain the advantage of heterostructures based on ZnO, TiO2 and SnO2.

Lines 59-62

Q4. While it is true that lack of standardization impedes a comparison with the literature, it is still of interest to show corresponding results for the chosen oxide photocatalysts. Obviously, the photocatalytic performance depends on the radiation wavelength, photon flux.

A4. This is the reason why we avoid comparing results. The papers contain different and sometimes insufficient information’s.  However we have followed your advice and the Introduction part was upgraded with additional explanations.

Lines: 46 – 55

Q5. The authors fail in supporting the choice of the pollutant types investigated.

A5. We have inserted new sentences to explain why acetamiprid and tartrazine where chosen to evaluate the photocatalytic activity of the tandem structure.

Lines 64-70

Q6. Why was Doctor Blade selected to obtain the tandem structures?

A6. We have also used other techniques (robotic spray pyrolysis, dip coating) to obtain tandem structures. However, doctor blade has several advantages such as: easy to employed science doesn’t require sophisticated equipment, good reproducibility or possibility to easy alternate the materials during the film deposition. However, there are also disadvantages due to high quantity of residual materials, possible contamination with organic additives and necessity of post-deposition annealing treatments.

Q7. The section 2.1. could be compressed as the synthesis details are similar. Doctor Blade details are missing.

A7. We have inserted new informations containing details on the deposition parameters.

Lines 77-92.

Q8. English needs revision (e.g. L79, L286-287)

A8. The English language andand typo mistakes were revised through the entire manuscript.

Q9. Describe the method employed to obtain the UV-VIS calibration curve in section 2.2.

A9. Thank you for the sugestion. We have inserted additional information regarding the procedure for UV-Vis calibration curve.

Lines 129-133

Q10. It is advised to show also the first layer morphology.

A10. Based on your sugestion we have inserted new figures presenting the SEM investigations of ZnO, SnO and TiO2 films.

Lines 198-205

Figures 2a, 2b and 2c

Table 3

Reviewer 3 Report

In this manuscript the author is suggesting couple of tandem structures for photocatalytic removal of two organic pollutants. The manuscript is written fine, but it needs some major and minor corrections. If the author could address the major corrections, it is fine for the publication otherwise I do not recommend it for publication at the present form.

Major:

1- What is the possibility of the agglomeration of the similar particles in the mentioned fabrication process? How one can be sure that different materials are uniformly mixed in the preparation step.  From the SEM images the surface of the tandem layer does not look so uniform and smooth. I do not recommend to compare the current samples to the tandem structures from the previous reports where different fabrication methods were used. And apply the same theoretical methods to defer the band structure of the tandem layer in this work. The author must use a characterisation method such as UV-VIS spectroscopy to understand the band structure of the tandem structures which are fabricated for this work.

2- The author claim that adjusting the mass ratio between the two semiconductor material would improve the photocatalytic effect of the final tandem structure. The author also experimentally showed that the photocatalytic effect of the sample which contains more TiO2 or ZnO is more promising under the UV irradiation. This is a clear fact that TiO2 and ZnO are fantastic inorganic UV absorbers, so the conclusion about the samples which contains more of these materials is out of discussion. To prove the importance of the suggested tandem structure in this manuscript, the author MUST compare the photocatalytic activity of the tandem structure to samples which only contains TiO2 and ZnO. However, in this manuscript only tandem structures are compared to each other.  

Minor:

3- The Manuscript need some minor language editing. The typos are plenty, and I suggest not to use the auto-correction for the editing!

4- I suggest to re-order the materials and methods section in a way that it matches the results in the manuscript.

5- There is one reference missing from line 94. If there is any?

6- The author should clarify what kind of particles where used in this work. Nanoparticles? Micron-size?

Author Response

Dear Reviewer,

We express our gratitude to your work and guidance that was helping us to improve the quality of this manuscript.

We have considered all your comments and suggestion in the new revised form of the manuscript. The changes where highlighted in red.

In this manuscript the author is suggesting couple of tandem structures for photocatalytic removal of two organic pollutants. The manuscript is written fine, but it needs some major and minor corrections. If the author could address the major corrections, it is fine for the publication otherwise I do not recommend it for publication at the present form.

Major:

Q1. What is the possibility of the agglomeration of the similar particles in the mentioned fabrication process? How one can be sure that different materials are uniformly mixed in the preparation step.  From the SEM images the surface of the tandem layer does not look so uniform and smooth. I do not recommend to compare the current samples to the tandem structures from the previous reports where different fabrication methods were used. And apply the same theoretical methods to defer the band structure of the tandem layer in this work. The author must use a characterisation method such as UV-VIS spectroscopy to understand the band structure of the tandem structures which are fabricated for this work.

A1. Thank you for the comments. I made several changes into the manuscript in order to improve the discussions regarding the experimental results. The aggregates formation is a certain fact in this kind of deposition procedures. The particles tend to form agglomerations not only in the same material composition but also between different materials composition. This is way we have inserted new SEM images showing the morphology of single component material (ZnO, TiO2, SnO2). Regarding the film homogeneity we give more explanation regarding the deposition technique. It is clear that the film composition is not exactly the same in each point. This is why we have indicate in the EDX results that the surface composition presents some abatements comparing with the initial precursor mass ratio composition, but still keeping the overall values. Considering the band gap evaluation, we totally agree with the reviewer and we used the UV-Vis spectroscopy to make the sample characterizations. All the results are adapted to these particular samples and the comparison with other reported data are used in scientific proposes.

Lines 177-181

Lines 198-205

Table 3

Figures 2a, 2b, 2c

Q2. The author claim that adjusting the mass ratio between the two semiconductor material would improve the photocatalytic effect of the final tandem structure. The author also experimentally showed that the photocatalytic effect of the sample which contains more TiO2 or ZnO is more promising under the UV irradiation. This is a clear fact that TiO2 and ZnO are fantastic inorganic UV absorbers, so the conclusion about the samples which contains more of these materials is out of discussion. To prove the importance of the suggested tandem structure in this manuscript, the author MUST compare the photocatalytic activity of the tandem structure to samples which only contains TiO2 and ZnO. However, in this manuscript only tandem structures are compared to each other.  

A2. Based on your recommendation we have inserted the bare ZnO, SnO2 and TiO2 in the photocatalytic experimental part. All graph and tables regarding the photocatalytic activity were changed in order to include three new samples. The discussions were upgraded to include the new experiment. It is true that TiO2, SnO2 and ZnO take individually are materials which use mostly the UV light spectra. When they are combined in tandem or heterostructures systems the optical properties may change due to the effective band gap of the type II mechanism. Scanlon was the first (from my knowledge) who made and reported this observation based on his work regarding photoactive heterostructures materials. We made several investigations on different material and the results indicate that the effective band gap may change in certain conditions (most of them depends on the tandem composition and crystalline structure). The tandem system will still absorb mostly in UV region but also can use a small part of Vis region (at the boundary between UV-Vis).

Lines 229-233

Lines 239-244

Lines 250-255

Lines 356-361

Figures 2, 3, 4 and 5

Tables 3 and 4

Minor:

Q3. The Manuscript need some minor language editing. The typos are plenty, and I suggest not to use the auto-correction for the editing!

A3. Thank you for pointing this typo mistake. We have corrected the title including the tandem structure composition.

Q4. I suggest to re-order the materials and methods section in a way that it matches the results in the manuscript.

A4. We made changes in both Methods and Discussions chapters to make this manuscript easier to follow.

Q5. There is one reference missing from line 94. If there is any?

A5. We have inserted a supplementary reference in this case.

Reference 52

Q6. The author should clarify what kind of particles where used in this work. Nanoparticles? Micron-size?

A6. Thank you for the suggestion, we have included into the manuscript the particle sizes (as provided by the producer company).

Lines 198-200

Reviewer 4 Report

This work focuses on the use of tandem Photocatalytic layers [ZnO and SnO2] against the photodegradation of acetamiprid pesticide (Apd) and tartrazine dye (Tr).

This is indeed an interesting work. 

The authors have characterized their samples using XRD and SEM, while they have checked their photocatalytic efficiencies and kinetics. Moreover, they have calculated the energy gap of their samples to support their findings.

I feel some issues have to be resolved:

-In the introduction part a few more information is needed, regarding the use of thin and thick metal oxide films for photocatalytic applications. More citations are needed.

-Why do we need the tandem photocatalysts? The authors should analyze more about the effect of the tandem photocatalysts in the introduction part (with references etc.)

-we need to see the photocatalytic efficiency of bare ZnO and bare SnO2 layers along with the tandem devices. Include such data in the existing graphs.

-what is the thickness of each layer (ZnO and SnO2, respectively)?

-What is the roughness of the films? AFM images are needed.

-the authors should give more information about acetamiprid pesticide (Apd) and tartrazine dye (Tr).

-the authors should give more information about the Photocatalytic process. I guess they used UV-vis spectroscopy, they had the absorption spectra, used either the max intensity or the integrated area, normalized the data etc. Please include this kind of information. 

-the authors should check the re-use of their samples at least for 3 to 5 times. This is essential if we want to talk about real life applications.

-what is the stability and adhesion of the samples?

This manuscript could be published after taken care of the above minor issues. 

Author Response

Dear Reviewer,

We express our gratitude to your work and guidance that was helping us to improve the quality of this manuscript.

We have considered all your comments and suggestion in the new revised form of the manuscript. The changes where highlighted in red.

This work focuses on the use of tandem Photocatalytic layers [ZnO and SnO2] against the photodegradation of acetamiprid pesticide (Apd) and tartrazine dye (Tr).

This is indeed an interesting work. 

The authors have characterized their samples using XRD and SEM, while they have checked their photocatalytic efficiencies and kinetics. Moreover, they have calculated the energy gap of their samples to support their findings.

I feel some issues have to be resolved:

Q1. In the introduction part a few more information is needed, regarding the use of thin and thick metal oxide films for photocatalytic applications. More citations are needed.

A1. The Introduction part was significantly improved in order to underline the importance of metal oxide films in photocatalytic applications. Additional references were inserted.

Lines 46-56

Lines 60-63

Lines 65-71

References 41-51

Q2. Why do we need the tandem photocatalysts? The authors should analyze more about the effect of the tandem photocatalysts in the introduction part (with references etc.)

A2. Thank you for the suggestion. We have inserted several examples of tandem photocatalysts and working conditions (including the references).

Lines 46-56

References 41-44

Q3. We need to see the photocatalytic efficiency of bare ZnO and bare SnO2 layers along with the tandem devices. Include such data in the existing graphs.

A3. Based on your recommendation we have inserted the bare ZnO, SnO2 and TiO2 in the photocatalytic experimental part. All graph and tables regarding the photocatalytic activity were changed in order to include three new samples. The discussions were upgraded to include the new experiment.

Lines 229-233

Lines 239-244

Lines 250-255

Lines 356-361

Figures 2, 3, 4 and 5

Tables 3 and 4

Q4. What is the thickness of each layer (ZnO and SnO2, respectively)?

A4. We have prepared and evaluated the bare ZnO, SnO2 and TiO2 thickness and the results are included in Table 3.

Q5. What is the roughness of the films? AFM images are needed.

A5. Thank you for the suggestion. Unfortunately, with our AFM device (NT-MDT) it was not possible to evaluate the RMA due to surface non-uniformities. The values obtained were not consequent through the entire surface.

Q6. The authors should give more information about acetamiprid pesticide (Apd) and tartrazine dye (Tr).

A6. We have inserted new sentences to explain the reasons of choosing these two substances.

Lines 65-71

Q7. The authors should give more information about the Photocatalytic process. I guess they used UV-vis spectroscopy, they had the absorption spectra, used either the max intensity or the integrated area, normalized the data etc. Please include this kind of information. 

A7. We have explained in more details how the photocatalytic activity was evaluated.

Lines 130-134

Q8. the authors should check the re-use of their samples at least for 3 to 5 times. This is essential if we want to talk about real life applications.

Q9. What is the stability and adhesion of the samples?

A8 and A9. We will give a comprehensive answer to both questions. Long time experiments and photocatalysts stability (chemical and mechanical) are indeed a pre-requisite before the transfer at large application scale. We have established the procedures required for these tests and the results will be the subject of future article. We have decided, based on the amount of experimental data, to split this work in two parts. One, which is the subject of this paper, is to correlate the photocatalytic activity with other parameters (composition, radiation, pollutant type). The second part will presents long time experiments and chemical/mechanical stability. Until now, using maximum 14h experiments the photocatalytic activity presents only small variation and the film integrity was preserved.

Round 2

Reviewer 1 Report

The authors have modified the manuscript satisfactorily according to the referees' guides.

Reviewer 3 Report

I see some typos here and there which I hope they would be corrected in the final version. I pass on this one. Good luck!